# Methods of Respiratory Virus Detection: Advances towards Point-of-Care for Early Intervention

**DOI:** 10.3390/mi12060697

**Published:** 2021-06-15

**Authors:** Siming Lu, Sha Lin, Hongrui Zhang, Liguo Liang, Shien Shen

**Affiliations:** 1Department of Clinical Laboratory, Zhejiang Hospital, Hangzhou 310003, China; lusiming@zju.edu.cn (S.L.); zhang2821785@163.com (H.Z.); 2Department of Laboratory Medicine, The First Affiliated Hospital, College of Medicine, Zhejiang University, Hangzhou 310003, China; 1508092@zju.edu.cn; 3Centre for Clinical Laboratory, The First Affiliated Hospital of Zhejiang Chinese Medical University, 54 Youdian Road, Hangzhou 310006, China

**Keywords:** COVID-19, point-of-care, respiratory virus, detection

## Abstract

Respiratory viral infections threaten human life and inflict an enormous healthcare burden worldwide. Frequent monitoring of viral antibodies and viral load can effectively help to control the spread of the virus and make timely interventions. However, current methods for detecting viral load require dedicated personnel and are time-consuming. Additionally, COVID-19 detection is generally relied on an automated PCR analyzer, which is highly instrument-dependent and expensive. As such, emerging technologies in the development of respiratory viral load assays for point-of-care (POC) testing are urgently needed for viral screening. Recent advances in loop-mediated isothermal amplification (LAMP), biosensors, nanotechnology-based paper strips and microfluidics offer new strategies to develop a rapid, low-cost, and user-friendly respiratory viral monitoring platform. In this review, we summarized the traditional methods in respiratory virus detection and present the state-of-art technologies in the monitoring of respiratory virus at POC.

## 1. Introduction

Viral infections of the respiratory tract have been threatening human life and well-being with an increasingly stronger effect from the outbreak of SARS in China in 2002, the global appearance of influenza A (H1N1) in 2009, to the novel coronavirus disease (COVID-19) [1,2,3]. In addition, respiratory infectious diseases on a massive scale led to a huge economic burden and humanitarian disaster to all nations affected. For example, to fight the further spread of COVID-19, US$675 million worth of preparatory work was launched from February through April 2020 [4]. Moreover, the world’s economic output fell by US$50 billion in 2003 due to life saving measures [5]. On the other hand, the rapid spread of infectious respiratory diseases evidently prompts significant steps toward disease prevention and easing economic pressure. For example, the basic reproduction number (R0 value) of the SARS-CoV-2 was 2.0 in the early stage, which is much greater than that of SARS in 2003 (1.7 to 1.9) [6]. Consequently, rapid and effective means of monitoring can alleviate the spread of such diseases, reduce the accompanying economic burden, and gain valuable time toward effective disease prevention.

Currently, there are two approaches to monitor respiratory viruses, i.e., measuring antibodies in the blood plasma produced in response to viruses (IgG & IgM), which reflects the functionality of the host immune system, and detecting viral load in respiratory specimens, which indicates viral replication in the primary infection site of individuals [7,8,9]. Viral load is generally measured using commercial RNA assays, such as the GeneXpert Dx System; IgG and IgM are monitored using a chemiluminescence immunoassay [10,11]. Nevertheless, the implementation of these assays requires high-cost equipment (e.g., thermal cyclers), highly skilled personnel and expensive reagents; thus, they are unsuitable for resource-limited settings. Moreover, these techniques are time-consuming, thereby delaying effective treatment and leading to the lack of ability to control the spread of infectious diseases in a timely manner. In particular, the recent occurrence of COVID-19 has caused a large-scale pandemic even affecting developed countries, and the existing RNA assays could no longer meet large-scale screening requirements during the outbreak period [12]. At the same time, in backward regions, the lack of resources and time-consuming nature of detection led to great challenges in focusing on the prevention and control of infectious diseases. Therefore, there is an urgent need for viral load detection to monitor the virus and thus provide information to manage infected patients in a timely manner.

Herein, we review emerging technologies in the development of respiratory viral load assays for point-of-care (POC) testing. Firstly, we present traditional assays for respiratory viral load monitoring, and then discuss existing improvements to these assays. We also elaborate on the commercial rapid PCR detection platform and immunoassays, portable amplification systems, and upcoming new approaches such as biosensors and microfluidics-based virus detection that have been proposed for POC viral load detection.

## 2. Conventional Respiratory Virus Quantification Methods

The traditional approaches for diagnosing infectious respiratory viral diseases are based on virus-infected cell cultures, nucleic acids or viral antigens and antibodies, which require trained operators, bulky instruments and have a significant time requirement. Moreover, methods based on electron microscopy, cytology or sequencing are limited in clinical laboratories. A brief overview of the quantification methods of respiratory viruses are presented in the following sections (see Table 1).

Virus isolation and culture is still considered as the gold standard for the laboratory-based identification of respiratory viruses [13]. After the successful isolation of viruses, electron microscopy is required to confirm their presence and immunofluorescence staining for their identification [14]. A further method for virus detection is hemadsorption, which is cell culture-based [15]. In this method, the erythrocytes adsorb to the plasma membrane of the virus-infected cell monolayer and aggregate, which can be subsequently detected by the naked eye. Cell culturing is a highly accurate standard technique for virus detection, while its four-week time requirement is too long to be applied during a potential viral disease outbreak. Aiming to speed up the process of virus isolation, the emergence of rapid virus culture methods, such as centrifugal enhancement technology (shell vial method), have significantly shortened the test cycle from one week to one to two days [16]. The shell vial method has greatly enhanced the isolation efficiency of many viruses, including the influenza virus, dengue virus and several respiratory viruses [16,17,18].

Since most respiratory viruses are very small and cannot be observed directly under a light microscope, the only method to detect them directly is transmission electron microscopy (TEM) [19,20,21]. Although TEM is being replaced by more sensitive methods such as PCR and immunofluorescence assay, it still plays an important role in some aspects of virology research, such as the discovery of new viruses, virus characterization, and titer determination. A major advantage of TEM is that it does not require virus-specific detection reagents. This is especially important in the case of a large-scale outbreak of an unknown viral pathogen, since targeted detection reagents and methods would not be available for an unknown type of pathogenic virus. For example, negative staining TEM technology has proven to be an essential tool for the discovery and identification of new viruses such as Ebola or SARS viruses [14,22,23]. Another example is COVID-19, in the early stages of which scientists captured the electron micrograph of the virus and determined its morphological properties, thus saving valuable time for prevention and control [24]. However, due to the high operational cost of the instrument and the need for considerable space and connecting facilities, TEM analysis can only be carried out in certain qualified centers.

In the clinical aspect, the routine methods for detecting respiratory viruses are immunology-related (e.g., radio-immunoassay (RIA), enzyme-linked techniques (EIA)), which are based on the reaction of antigen and antibody [25]. RIA utilizes radioisotope-labeled antibodies/antigens to detect antigens/antibodies [26]; however, owing to the associated radiation risk, the enzyme immunoassay (EIA or ELISA) was developed as an alternative method. This and the combination of immunochemical reactions with a chemiluminescence technique (chemiluminescence immunoassay or CLIA) feature high sensitivity and thus have been considered routine assays for protein detection in the clinical laboratory practice [27]. In particular, ELISA, which can be used to measure proteins at extremely low levels (10^−12^~10^−9^ mol/L), has been utilized in influenza virus detection [25]. Currently, chemiluminescence-based serological methods are common in clinical laboratories and rely on bulky automated machines. Following a respiratory virus infection, the immune system reacts to the virus and produces antibodies. The detection of the levels of these antibodies can monitor the degree of response to viral antigen exposure by the immune system, including IgM responses that represent acute infections, and IgG produced as a result of primary infection or in the acute phase of a secondary infection. Nevertheless, immunoassays have the disadvantage of producing false negative results during the time window between the viral infection and the start of antibody production.

Nucleic acid detection-based technology is a highly accurate method for the clinical diagnosis of respiratory viral load, which has revolutionized respiratory virus-related diagnostics by eliminating the false negative window effect [28]. This method has high sensitivity, good stability, and a shorter reaction time compared to traditional culture methods. Nucleic acid expansion can allow for the amplification of a specific region of a DNA sequence 10^6^-fold in vitro, and also permits the reverse-transcription of RNA into DNA and subsequent execution of PCR analysis. Moreover, the real-time RT-PCR assay for detecting influenza virus is considerably faster than the endpoint detection method, and its sensitivity is comparable to or better than that of cell culture methods that are considered as reference methods for viral detection [29]. More recently, based on nucleic acid detection, the cobas^®^ SARS-CoV-2 qualitative assay was developed for detecting COVID-19 infections, which even obtained FDA emergency use authorization [30]. The cobas^®^ SARS-CoV-2 is a real-time RT-PCR test for the cobas^®^ 6800/8800 system designed to qualitatively evaluate nasopharyngeal and oropharyngeal swab samples from patients who meet the clinical or epidemiological criteria for COVID-19 nucleic acids in SARS-CoV-2. The 6800 version of this system can detect 1440 samples per day, whereas the 8800 version has a performance of 4128 samples. Compared with other existing techniques, this nucleic acid amplification system improves the coronavirus detection time by 10-fold, thus earning precious time for disease prevention. Although nucleic acid testing provides accurate results for respiratory viruses, this method requires a rigorous laboratory infrastructure and specially trained clinical laboratory operators.

## 3. Commercialized Respiratory Virus Diagnostic POCT Devices

With the aim to provide viral detection during an outbreak, inexpensive alternatives have been developed and commercialized (Table 2), such as the GeneXpert Dx System [31], Luminex xTAG^®^ RVP [32], BinaxNOW^®^ Influenza A&B or Clearview^®^ Exact Influenza A&B [33]. Therefore, serological antibody testing is important to verify the effectiveness of virus treatment [34]. The GeneXpert Dx System is based on the multiplex real-time reverse transcriptase polymerase chain reaction (RT-PCR) to detect the influenza A and B matrix proteins [31]. The Luminex xTAG^®^, also based on PCR technology, measures 18 respiratory viruses and subtypes in a short time period [32]. In this system, the amplified DNA is mixed with short-sequence TAG primers to recognize each viral target. Once the target is present, the target-specific primers are extended and incorporated simultaneously with the label. These alternative methods have significantly reduced the cost of testing compared to traditional viral load assays. Both the BinaxNOW^®^ Influenza A&B and Clearview^®^ Exact are based on immunochromatography technology that measures the viral nucleoprotein antigens [33,34]. These methods can rapidly produce results in 15 min. On the downside, they both have insufficient sensitivity, and are prone to false negatives. Therefore, their test results need to be combined with clinical practice to make accurate diagnoses.

### 3.1. Molecular Diagnostics

The GeneXpert Dx System is a PCR-based assay to measure several respiratory viruses at the same time, such as influenza A/B, and respiratory syncytial virus [31]. It is an automated instrument combined with a barcode scanner, a laptop, and a processing unit. The latter carries several instrument modules, and each module performs the PCR reaction of the sample. Moreover, the sample does not need pretreatment, and it takes only 30 min from sample preparation to produce an output. To verify the reliability of this system in clinical practice, Ling et al. compared the Xpert Xpress Flu/RSV test with two other conventional molecular methods [35]. This kit showed 100% specificity and sensitivity for influenza A and RSV, and 100% sensitivity and 97.8% specificity for influenza B. Moreover, the results of Xpert Xpress Flu/RSV and the results of the other two systems showed a high degree of consistency. It is worth noting that the speed of the GeneXpert Xpress was up to fivefold faster than the other two for batch test runs of 12 samples, which could make faster treatment decisions.

Recently, as COVID-19 has greatly dispersed around the world, Xpert^®^ Xpress SARS-CoV-2 was fabricated for the rapid detection of SARS-CoV-2, which has been demonstrated to be a highly-sensitive test in numerous analytical and clinical studies [36,37]. This kit provides the speedy detection of this virus in 45 min, and it only takes one minute to complete the sample preparation. Furthermore, this kit showed sensitivity of 100% with a limit of detection (LOD) ranging from 8.3 to 60 cp/mL for SARS-CoV-2 detection. As a rapid detection device, this kit showed excellent agreement with the Roche Cobas 6800 system and traditional RT-qPCR. The Xpert^®^ Xpress SARS-CoV-2 test meets the current urgent need for COVID-19 testing, allowing clinicians to promptly evaluate patients receiving treatment in healthcare facilities. Accurate testing around patients is revolutionary, helping to alleviate the pressure on medical institutions caused by the outbreak of COVID-19, so that they can allocate respiratory isolation resources rationally.

### 3.2. Immunology-Based Rapid Detection Assay

The Alere BinaxNOW^®^ Influenza A&B kit is an in vitro immunochromatographic reagent for the detection of influenza virus nucleoprotein antigens in nasopharyngeal swab samples. It can quickly distinguish between A and B virus infections. For example, the comparison of test results between a traditional RT-PCR reagent and the Alere BinaxNOW^®^ Influenza A & B kit used for nasal wash specimens from 240 pediatric patients showed that the detection rate of two influenza viruses for Binax was greater than 97.9%, and the sensitivity for both was almost the same as that of RT-PCR [38]. Another study showed that the BinaxNOW rapid test has high sensitivity and accuracy for the detection of influenza A virus at any age in the acute stage of the outbreak and in patients under 18 years of age [39].

Immunology-based rapid diagnostic tests are not restricted to the detection of respiratory virus antigens but can also detect virus-specific antibodies. Recently, researchers found that, in COVID-19 infection, the antibody level detected by EIA is important for vaccine studies, and the use of convalescent plasma or therapeutic monoclonal antibodies could either improve the clinical outcome or cause immunopathological damage to the recipient [40]. Therefore, serological antibody testing is important to verify the effectiveness of virus treatment. A device such as COVID-19 IgM/IgG Rapid Test from BioMedomics is a good example [41], which has the ability to screen symptomatic or asymptomatic virus carriers in a short period of around 10 to 15 minutes.

## 4. Advances in Respiratory Virus Testing for Rapid Diagnosis

During a virus outbreak, rapid detection of respiratory viral load is essential to facilitate decision-making on the initiation of anti-respiratory viral treatment (ART) or the identification of virological ART failure. Although inexpensive viral load assays can be measured in central laboratories, they cannot accommodate the huge number of samples during a virus outbreak, and are unable to carry out tests in remote areas, such as those with a lack of negative pressure environment, PCR detectors, etc. Aiming to address this challenge, one option is to store the samples in a special transfer box and send it to the designated testing unit. However, this not only lengthens the turnaround time from sample to answer, but also has the risk of infectious virus leakage during the change of plan process. Ideally, medical workers can use systems that do not require instrumentation or are portable and can read data in less than 30 min to facilitate clinical decision and thus limit viral spread. Aiming to address these challenges, new technologies based on nanotechnology and microfluidics have been developed toward respiratory viral detection in a POC testing. These include rapid molecular diagnosis (LAMP), paper-based technology, biosensors, microfluidics, and smartphone-based imaging technologies.

### 4.1. Rapid Molecular Diagnosis

Loop-mediated isothermal amplification (LAMP) is a rapid molecular diagnosis based on a novel nucleic acid amplification method, initially developed by Notomi et al. [42]. It relies on four primers (two outer primers and two inner primers) and a strand displacement DNA polymerase. The four primers can recognize six specific fragments of conserved-sequence DNA. The main principle of LAMP technology is that DNA can be in a dynamic equilibrium state at about 65 °C. At this temperature, four specific primers and a strand displacement DNA polymerase are utilized to synthesize DNA by self-circulating. Based on this method, several infection kits have been developed for POC. For example, in November 2020, FDA issued an emergency authorization to approve the Lucira Covid-19 All-in-One Test Kit—the first home self-test kit that could provide rapid results [43]. The kit utilizes LAMP technology to rapidly produce results within 30 min and has a 100% conformance rate compared to highly sensitive detection methods. Furthermore, based on the characteristics of LAMP and using its principle, many studies have been dedicated to the development of rapid detection methods for respiratory viruses.

Recently, researchers have developed LAMP-based novel assays to detect respiratory viruses, such as influenza viruses, MERS-CoV, and SARS-CoV, for POC testing. For example, a rapid and colorimetric multiplex reverse transcription LAMP (RT-LAMP) method was developed for influenza viruses (e.g., H5N1, H5N6, H1N1, H3N2 and H7N9) [44]. Compared to conventional PCR methods, which need around two to three hours for the full reaction and observation, this novel method only requires one hour for several target genes (i.e., H1N1, H3N2, H5N1, H5N6, H5N8 and H7N9) and without additional steps. Moreover, the RT-LAMP assay shows specificity to infectious viruses and capability to intuitively detect these viruses by a one-pot colorimetric visualization assay, making it a more feasible POC test. This test showed a high sensitivity of 100 to 0.1 viral genome copies. In another study, a nucleic acid visualization method was developed for detecting MERS-CoV nucleic acids, which is based on a vertical-flow colloidal gold particle binding strip (Figure 1) [45]. This novel visualization method shows significant advantages over existing MERS-CoV rapid detection assays, since two loop primers (LF and LB) are labeled with FITC or biotin to provide visible results. This assay was shown to be capable of detecting 1 × 10^1^ copies/µL of MERS-CoV RNA, which is regarded as more sensitive than a conventional RT-LAMP assay. In addition, it showed high specificity and exhibited no cross-reactivity with any of SARS-CoV, HKU4, HKU1, OC43 or 229E. Considering the equipment-free nature and rapid result output, the visualized strip is a promising approach for the management of MERS at the POC approach. Focusing on the currently circulating SARS-CoV-2, LAMP is expected to play an important role in its detection, so that infected persons will be localized as soon as possible for isolation to avoid the further spread of the virus and thus slow down the outbreak.

### 4.2. Immunoassay Paper-Based Devices/Technology

In recent years, owing to the superior performance of paper-based technology in POC testing, such as affordability, user-friendliness, speed, robustness, and scalability, it has been utilized to create inexpensive diagnostic tools to improve global health. Moreover, due to capillary action, liquid flow in paper does not require any external power, providing paper, as a base material, excellent potential for the construction of a portable, integrated, human-readable diagnostic device.

A lateral flow test strip is a type of paper-based device/technology used for the detection of respiratory viruses. Various particles, such as gold nanoparticles (AuNPs), have been increasingly employed in the development of lateral flow test strips for their unique optical, electronic, and/or structural properties. For example, Das et al. reported a rapid and sensitive determination of the total iron-binding capacity of transferrin (Tf) in human serum using surface-enhanced Raman scattering (SERS) spectroscopy [46]. In the study, metal organic framework (MOF)–gold nanoparticle (AuNP) complexes were used as the SERS substrate. This SERS-based analysis of Tf using MOF–AuNPs provides new insight for the rapid and sensitive diagnosis of iron deficiency in human serum. For example, as a stabilizing agent, Au/Ag-coated Fe_3_O_4_ magnetic NPs were utilized as magnetic SERS tags for developing a novel surface-enhanced Raman scattering (SERS)-based strip to detect H1N1 and HAdV (see Figure 2A) [47]. Therein, the Fe_3_O_4_@Ag magnetic tags were conjugated with dual-layer Raman dye molecules and captured viral antibodies, recognizing and enriching the target virus. Notably, this strategy allows the strip to be directly utilized for real samples. Based on the high-performance Fe3O4@Ag SERS, this assay showed a detection limit of 50 and 10 pfu/mL for H1N1 and HAdV, respectively, which were 2000-fold lower than the standard gold strip. Moreover, the SERS strip has the potential to be used for the rapid detection of other targets, such as coronavirus, in complex specimens. Recently, an AuNP-based lateral flow test was developed to detect IgM and IgG antibodies simultaneously against the SARS-CoV-2 virus [48]. This kit combined with IgG and IgM antibody test kit, which showed a specificity of 90.63% with a sensitivity of 88.66%. In addition, the test results of venous blood samples and finger blood samples show that the test results of different types of blood samples are consistent. Importantly, owing to enhanced sensitivity and better utility achieved, the IgM-IgG combined assay holds great promise for the rapid screening of both symptomatic and asymptomatic SARS-CoV-2 carriers at the POC.

The clinical diagnosis of respiratory viruses often requires quantitative measurements of proteins or nucleic acids. However, due to the variation in visual perception of color among end-users and under different lighting conditions, visual colorimetric measurements by the naked eye have been insufficient for quantitative purposes. Therefore, to achieve quantitative analysis in paper-based diagnostics, readout platforms can be used to record the color intensity. For example, quantum dot (QD)-based immunochromatographic strips that are integrated with a mini 3D-printed readout platform were used for the seasonal (autumnal) measurement of avian influenza A (H7N9) virus [50]. The mini 3D-printed readout platform allows the assessment of the result by the naked eye. Owing to the properties of QD, such as high quantum yields (QYs), tunable emission wavelengths, large molar extinction coefficients, strong photostability, and broad absorption cross-sections, this biosensor showed a detection limit of 0.0268 HAU. Moreover, this platform exhibited a reaction time of 15 min, and the total coincidence rate between this platform and real-time PCR was 98%. In a different study, a paper-based analytical device integrated with pyrrolidinyl peptide nucleic acid (acpcPNA)-induced nanoparticles was used to detect *Mycobacterium tuberculosis* (MTB), human papillomavirus (HPV) and middle East respiratory syndrome coronavirus (MERS-CoV) (Figure 2B) [49]. In this paper-based device, once the target DNA is present, the formation of the anionic DNA-acpcPNA double strand will lead to the dispersion of AGNP due to electrostatic repulsion, thus leading to detectable color change, by which the result can be determined. This device was demonstrated to detect MERS-CoV, MTB and HPV in the range of 20 to 1000 nM, 50 to 25,000 nM, and 20 to 25,000 nM, respectively. Since the nanoparticle AbNP enhances the sensitivity of the device, detection limits of 1.27 (MTB), 1.53 (MERS-CoV), and 1.03 nM (HPV) were reported. This newly developed multiplex colorimetric paper-based device has the capability for rapid screening and detection in infectious disease diagnostics.

### 4.3. Biosensors for Respiratory Virus Detection

In the last decade, optical and electrochemical sensors have been widely proposed for respiratory virus detection. These sensors, defined as biosensors in analytical chemistry, rely on a biomolecule for molecular recognition and a transducer for an observable output, which can be implemented at the POC for respiratory virus detection. In this section, we present the state-of-the-art biosensors for the prevention and monitoring of respiratory viruses.

Immune technology-based biosensors are currently being developed for respiratory virology testing. For instance, a novel electrochemical influenza A biosensor was recently developed for the measurement of N activity, which is one of the glycoproteins wrapped around the flu virus (Figure 3A) [51]. In this biosensor, a gold screen-printed electrode (AuSPE) and a graphene-Au hybrid nanocomposite were utilized to improve the properties of the biosensor. As a result, the biosensor detected the flu virus ranging from 10^−8^ to 10^−10^ U mL^−1^, with a detection limit of 10^−8^ U mL^−1^. Moreover, this developed biosensor has achieved very successful results in the detection of real influenza virus A (H9N2). A different study featured another sensor, which is as an electrochemical immunosensor for the MERS-CoV [52]. This biosensor was based on competitive analysis performed on a carbon electrode dielectrophoresis (DEP) array modified with gold nanoparticles to capture the recombinant spike protein (S1). In addition, owing to the utilization of AuNP modified carbon array electrodes, this sensor presented a sensitivity of 0.001 ng·mL^−1^. Furthermore, the detection limit using this biosensor was improved to 1.0 pg·mL^−1^ within 20 min in comparison with 1 ng·mL^−1^ in ELISA within one to two hours. More recently, an ultra-sensitive impedimetric biosensor for the detection of influenza A viruses was fabricated [53]. Therein, a three-electrode system with K_3_[Fe(CN)_6_] as an electrochemical probe was employed. The monoclonal antibodies are coated on the electrode surface to detect the presence of viral antigens, and subsequent changes in the electrode after the antigen-antibody reaction are measured. This sensor demonstrated a detection limit of 0.79 fM and a linear range of 0.18 f. to 0.18 nM. Owing to the development of nanotechnology, such immunobiosensors have clearly shown great potential for the determination of respiratory viruses.

On the downside, immunological biosensors cannot be used for accurate detection in the early stages of viral infection; thus, the application of molecular detection compared to sensors is superior in viral detection at early stages, which are characterized by a higher time requirement for prompt clinical diagnosis, treatment, disease prevention and control. For instance, an arch-shaped multiple-target sensing for the rapid identification of infectious pathogens was developed (Figure 3B) [54]. In this study, 50 bp long oligonucleotide primers at 5 μM concentration were utilized to enhance the sensitivity of amplification approaches, which also inhibited primer dimerization. In addition, a variety of infectious pathogens (such as MERSCoV, HCoV, EBOV and ZIKV) were used for diagnostic and identification tests on the platform, which had a high accuracy in determining all pathogen types. In addition, this multiple-target sensing platform has the ability to simultaneously detect MERS-CoV and HCoV in 20 min in 20 clinical specimens. Thus, this arch-shaped multiple-target sensing platform can provide a rapid detection of pathogens in various clinical applications.

### 4.4. Microfluidic Device for Respiratory Virus Detection

Miniaturized RNA assay chips have been designed to carry out PCR in a microfluidic device. Owing to the fast-mixing rate and large surface-to-volume ratio of the device, nucleic acid testing can be completed in a short time period with reduced reagent consumption. For instance, detection platforms with integrated colorimetry were developed for multiple respiratory viruses (i.e., influenza A and B virus and human adenoviruses) (Figure 4A) [55]. This chip integrates sample preprocessing, nucleic acid extraction and amplification to perform the whole test in less than an hour, which is much quicker than commercial methods (three to four hours). In addition, the measurement of 109 clinical samples demonstrated that this system has a high specificity (100%, confidence interval 94.9 to 100.0) and sensitivity (96%, confidence interval 78.1 to 99.9). The multi-respiratory virus screening microfluidic chip provides a key strategy for the future development of further respiratory virus detection methods. Recently, a portable microfluidic system that simultaneously diagnoses seven human coronaviruses was developed in response to the current outbreak of SARS-CoV-2 [56]. This chip, which is based on the LAMP method, has the ability to simultaneously detect SARS-CoV, MERS-CoV, SARS-CoV-2 and four other human coronaviruses (HCoV) (i.e., HCoV-OC43, HCoV-229E, HCoV-HKU1 and HCoV-NL63) within 40 min. Furthermore, compared to the “gold standard” polymerase chain reaction, this device showed 100% detection consistency in 54 real clinical samples. Thus, it could prove highly beneficial for diagnosis of SARS-CoV-2 during the COVID-19 pandemic. Recently, a high-throughput, multi-index nucleic acid isothermal amplification analyzer, named RTisochip™-W, was also developed to determine 19 common respiratory viruses, including SARS-CoV-2 (Figure 4B) [57]. This analyzer utilizes a centrifugal microfluidic chip, which can analyze all viruses within 90 min, and features a detection limit of ≤50 copies·μL^−1^, which is equivalent to that of conventional RT-PCR. Moreover, the RTisochip™-W system showed strong robustness, good repeatability, and high specificity. Based on testing 14 COVID-19-positive samples, 201 preclinical samples, 25 clinical diagnosis samples and 614 clinical samples of patients with respiratory tract infections or those of suspected patients, this system demonstrated a good agreement (98.5% coincidence rate) with the reference results.

### 4.5. Smartphone-Based Detection Technologies

In order to significantly reduce the mortality and morbidity caused by viral infection in resource-poor areas, a small and easily accessible intelligent detection system, such as a smartphone-based even detection platform would be highly valuable. To satisfy this need, Xia et al. developed a ZnO nanorod-based colorimetric avian influenza virus detection platform, which is combined with a smartphone (Figure 5A) [58]. In this platform, the 3D nanostructures are made on PDMS serving as a scaffold for conjugating antibodies. Once the virus is captured, it can be detected by the naked eye within 1.5 h through the colorimetric reaction of gold nanoparticles. This system has demonstrated a sensitivity of 2.7 × 10^4^ EID50/mL, which is an order of magnitude higher than the conventional fluorescence-based ELISA assay. Meanwhile, owing to the data processing capabilities of smartphone imaging systems, a detection limit of 8 × 10^3^ EID50·mL^−1^ can be achieved. More recently, a smartphone-integrated instant care system for detecting SARS-CoV-2 virus from nasal swabs was developed, which is the first utilization of smartphone-based LAMP detection of pathogens for POC in animal application (Figure 5B) [59]. This chip utilizes LAMP assay for the detection of five bacterial and viral pathogens that cause equine respiratory infectious diseases, and can be evaluated in 30 min. Most importantly, it can be integrated with a telemedicine platform, and by using mobile devices as a testing tool, it is convenient for epidemiological reports and sharing test results with physicians. In addition, it can be easily applied to the detection of RNA viruses by using the one-step RT-LAMP protocol, simply by adding reverse transcriptase to the LAMP reaction mixture without modifying the buffer or reaction conditions.

## 5. Potential Techniques for Future POC Diagnosis

In addition to the new technologies listed above, there are many promising ones with the potential to be adopted as future POC diagnosis tools for detection of respiratory viruses, such as digital PCR, surface-enhanced Raman scattering (SERS), 3D plasmonic-based chips. For example, rapid detection of low virus (SARS-CoV-2) RNA load was achieved by using a droplet digital PCR method [60]. In this study, a flat pipette head with an elliptical cross section was utilized to prepare monodisperse droplets. By the end of PCR reaction, the fluorescence generated by the droplets with virus (as low as 100 target copies) can be captured by mobile phone software, and the presence or absence of the virus can be identified by the naked eye or images. This platform demonstrated a LODs of 3.8 copies per 20 μL of sample with a dynamic range of 4 to 100 copies. Additionally, the ddPCR platform is shown to be inhibitor-resistant with spiked saliva samples, indicating that RNA extraction may not be necessary.

Surface-enhanced Raman scattering (SERS) was reported to improve the sensitivity of detection of influenza A/H1N1 virus [61]. In this study, the surface energy difference between a perfluoro decanethiol (PFDT) spacer and the Au layer was utilized to fabricate a 3D nano-popcorn plasmonic substrate, making gold nanoparticles self-assemble and generating multiple hotspots between neighboring particles. The decrease of Raman peak intensity resulting from the release of Cy3-labeled aptamer DNAs from nano-popcorn substrate surfaces via the interaction between the aptamer DNA and A/H1N1 virus was used to quantitate the influenza A/H1N1 virus. This method showed approximately three orders of magnitude more sensitivity than that determined by the ELISA assay.

The plasmonic model, which can enhance the performance of existing platforms by providing stable, real-time, highly sensitive and label-free analyte detection, has emerged as a key candidate for the development of next-generation diagnostic technologies to reduce the burden of infectious diseases [62,63]. For example, a dual-functional plasmonic biosensor combining the plasmonic photothermal (PPT) effect and localized surface plasmon resonance (LSPR) sensing transduction was fabricated for COVID-19 diagnosis [64]. In the biosensor, the gold nanoislands (AuNIs) functionalized with complementary DNA receptors can perform a sensitive detection of the selected sequences from severe acute respiratory syndrome coronavirus 2 (SARS-CoV-2) through nucleic acid hybridization. For better sensing performance, the thermoplasmonic heat was generated on the same AuNIs chip when illuminated at their plasmonic resonance frequency. The localized PPT heat was able to elevate the in-situ hybridization temperature and facilitate the accurate discrimination of two similar gene sequences. This biosensor exhibited a high sensitivity toward the selected SARS-CoV-2 sequences with a lower detection limit down to the concentration of 0.22 pM and allows precise detection of the specific target in a multigene mixture. These studies reveal the applicability of digital PCR, surface-enhanced Raman scattering (SERS), plasmonic-based biosensors and other technologies in nucleic acid testing and viral disease diagnosis and provide an effective idea for the rapid detection of respiratory viruses in the future.

## 6. Conclusions

The detection of respiratory viruses, especially the recent outbreak of COVID-19, almost exclusively depends on RT-PCR or general molecular testing. At the same time, immunological methods suitable for extensive screening, such as rapid antibody tests, cannot confirm the presence of the virus in the early stage of infection. At present, mature rapid tests for respiratory viruses can only detect a certain virus, such as Xpert^®^ Xpress Flu/RSV that can identify the two viruses within 30 min. Developing a highly accurate set of tests for multiple respiratory viruses takes time, as viruses mutate under certain circumstances. Despite multiple efforts to develop rapid, sensitive, and selective assays, commercial kits for the rapid detection of multiple respiratory viruses (including influenza, SARS, SARS-CoV-2) have not yet been established. Thus, it is imperative for researchers to continue to develop such platforms to monitor the true global infection dynamics in a timely manner, and thus be able to take effective and timely control measures. In the meantime, with the global outbreak of COVID-19 still underway, researchers should effectively apply new technologies (i.e., SERS, digital PCR, plasmonic-based biosensors) for rapid virus detection to improve detection sensitivity and avoid missing the detection of low-copy viral load.

## Figures and Tables

**Figure 1 micromachines-12-00697-f001:**
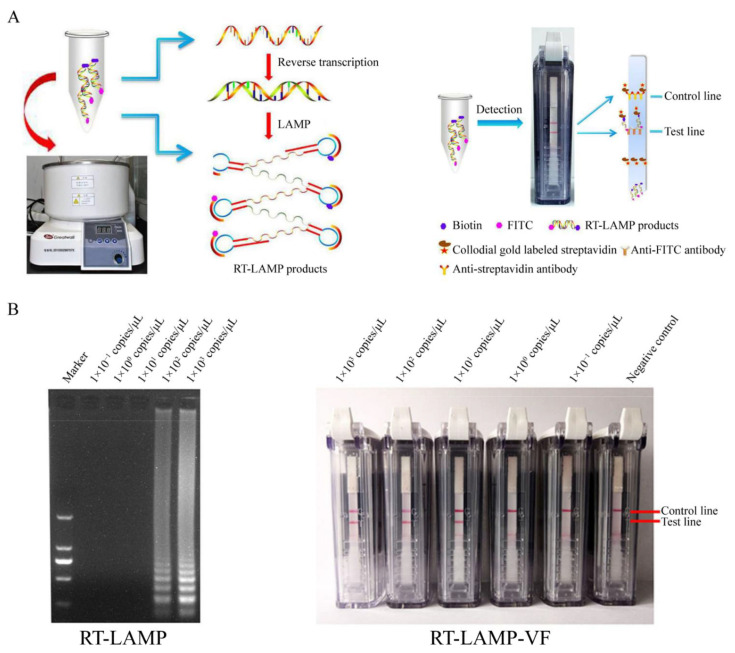
LAMP assay for MERS-CoV detection. (**A**) Schematic of the RT-LAMP with RT-LAMP-VF; (**B**) Comparison of sensitivity between conventional RT-LAMP and RT-LAMP-VF (reproduced with permission [45]. Copyright 2018, Frontiers). RT-LAMP: reverse transcription loop-mediated isothermal amplification; RT-LAMP-VF: RT-LAMP with a vertical flow visualization strip.

**Figure 2 micromachines-12-00697-f002:**
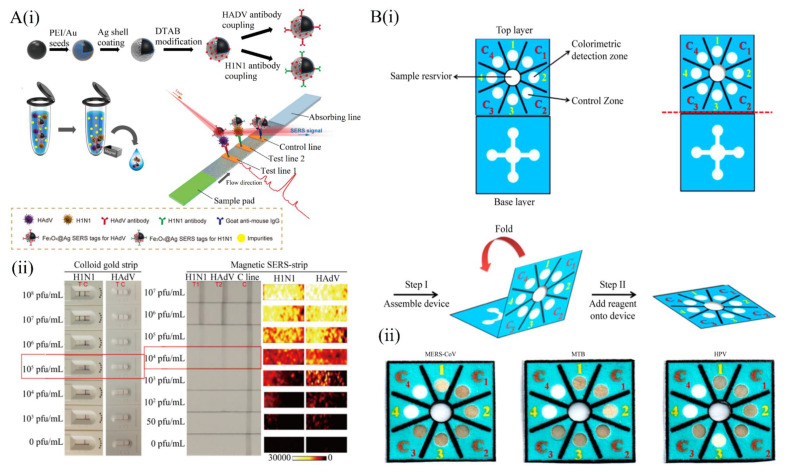
Paper-based strip for respiratory virus detection. (**A**) (**i**) Schematic of the Magnetic SERS Strip in detection of H1N1 and HAdV; (**ii**) Comparison of magnetic SERS strips and commercially gold strips for H1N1 and HAdV detection (reproduced with permission [47]. Copyright 2019, American Chemical Society). (**B**) (**i**) Schematic of Multiplex Paper-Based colorimetric device; (**ii**) selectivity of three virus (i.e., MERS-CoV, MTB, and HPV) detection via multiplex colorimetric paper device (reproduced with permission [49]. Copyright 2017, American Chemical Society).

**Figure 3 micromachines-12-00697-f003:**
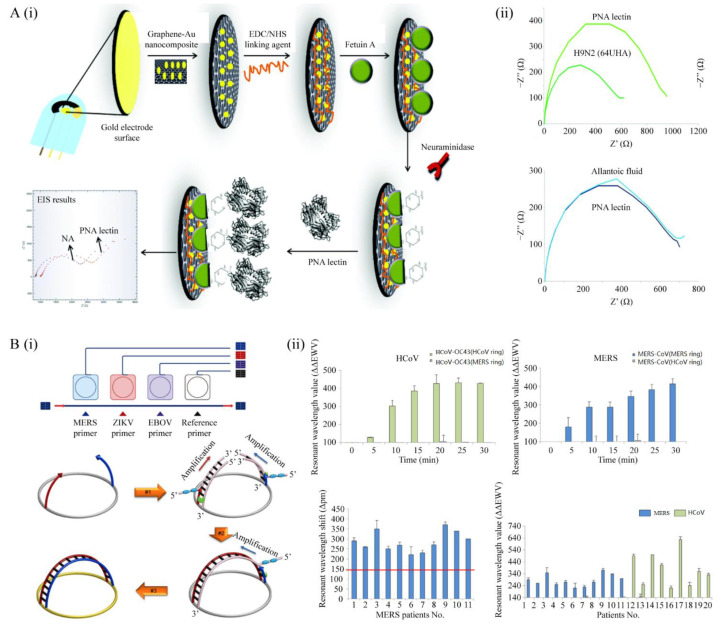
Biosensors for respiratory virus detection. (**A**) (**i**) Schematic of development of electrochemical influenza A biosensor; (**ii**) selectivity of the biosensor (reproduced with [51]. Copyright 2017, Royal Society of Chemistry). (**B**) (**i**) Scheme of the arch-shaped multiple-target sensing platform for diagnosis and identification of emerging infectious pathogens; (**ii**) utility of the arch-shaped multiple-target sensing platform in detecting of clinical samples (reproduced with [54]. Copyright 2018, Royal Society of Chemistry).

**Figure 4 micromachines-12-00697-f004:**
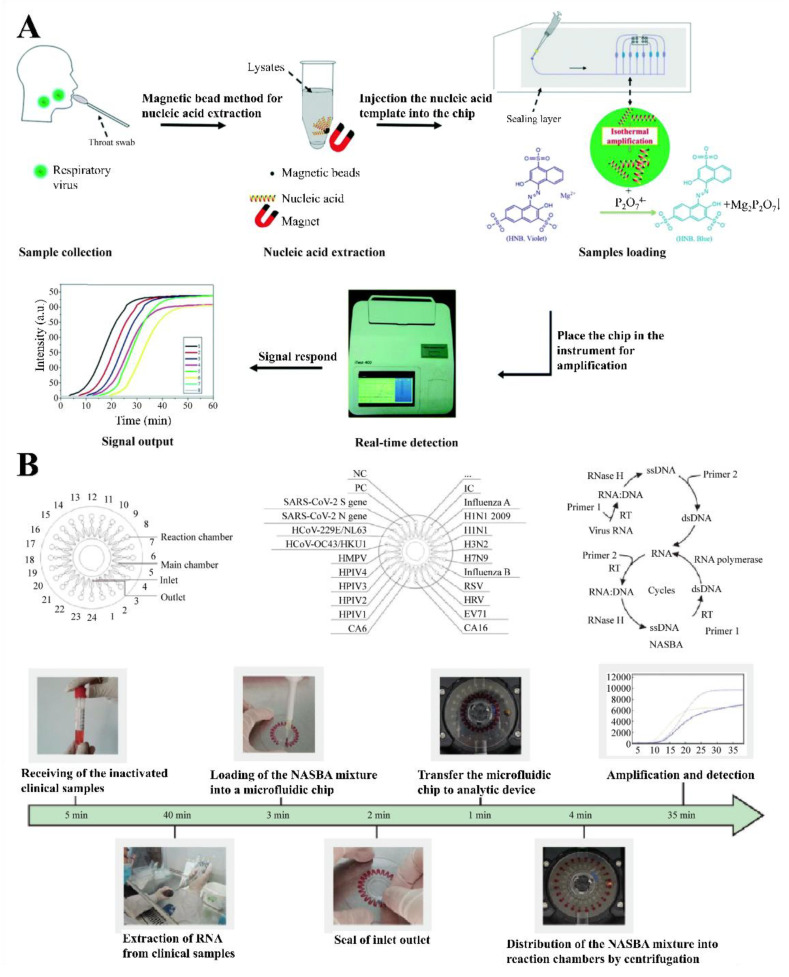
Microfluidic device for respiratory virus detection. (**A**) Schematic diagram of the LAMP-integrated microfluidic chip system for multiplexed respiratory virus detection (LMCS-MRVAs) (reproduced with [55]. Copyright 2018, Royal Society of Chemistry). (**B**) Schematic of the disc-shaped microfluidic device (reproduced with [57]. Copyright 2020, Elsevier).

**Figure 5 micromachines-12-00697-f005:**
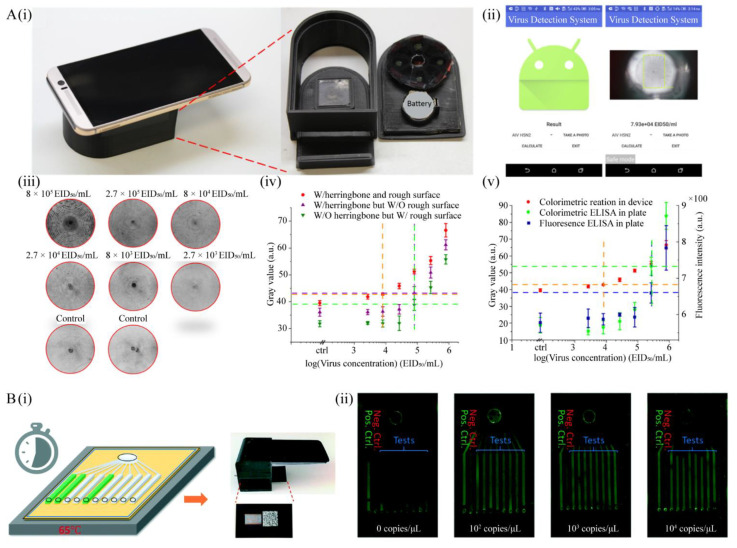
Smartphone-based POC platforms for respiratory virus detection. (**A**) (**i**) Photograph of the smartphone-based on-chip system; (**ii**) smartphone application interface; (**iii**) graphs of AIV virus detection with the smartphone-based on-chip detection system; (**iv**) comparison of with/without the herringbone structure and ZnO nanorod replication virus detection; (**v**) comparison of the colorimetric detection system and conventional ELISA assay in virus detection (reproduced with [58]. Copyright 2019, American Chemical Society). (**B**) (**i**) The chip is heated at 65 °C for LAMP reactions and inserted into the cradle for end-point fluorescence imaging; (**ii**) fluorescence image of EHV1 templates at different concentrations (reproduced with [59]. Copyright 2020, Royal Society of Chemistry).

**Table 1 micromachines-12-00697-t001:** Advantages and drawbacks of traditional virus detection methods.

Method	Principle	Time Required
Virus plaque test	infectivity detection	several days
Immunofluorescent plaque test	infectivity detection	several days
qPCR	viral nucleic acid detection	several hours
ELISA	viral protein detection	several hours
Hemagglutination assay	viral protein detection	several hours
Viral flow cytometry	viral particle detection	several hours
Transmission electron microscopy	viral particle detection	several weeks

**Table 2 micromachines-12-00697-t002:** Characteristics of commercial respiratory virus detection tests.

Company	Kit	Virus	Principle	Sample Type	Sensitivity	Specificity	Time
Abbott	SD BIOLINE Influenza Antigen	Influenza virus type A&B and H1N1	antigen test	human nasal swab, throat swab, nasopharyngeal swab or nasal/nasopharyngeal aspirate	91.8%	98.9%	5 min
Abbott	SD BIOLINE Influenza Ultra	Influenza virus type A&B	antigen test	nasopharyngeal swab, nasopharygeal aspirate	nasopharyngeal swab Flu A: 88.5% Flu B: 91.5% nasopharyngeal aspirate Flu A: 93.9% Flu B: 91.7%	nasopharyngeal swab Flu A: 97.4% Flu B: 97.4% nasopharyngeal aspirate Flu A: 97.7% Flu B: 97.7%	5–8 min
Abbott	CLEARVIEW^®^ EXACT INFLUENZA A&B	Influenza virus type A&B	antigen test	nasopharyngeal swab	Flu A: 81.7% Flu B: 88.6%	Flu A: 98.5% Flu B: 97.4%	15 min
Abbott	BINAXNOW^®^ INFLUENZA A&B	Influenza virus type A&B	chromatographic immunoassay	nasopharyngeal swab, nasopharygeal aspirate	Flu A: 70–89% Flu B: 50–69%	Flu A: 90–99% Flu B: 94–100%	15 min
BIOFIRE	The BioFire^®^ FilmArray^®^ Respiratory (RP&RP2) Panels	VIRUSES: Adenovirus; Coronavirus HKU1; Coronavirus NL63; Coronavirus 229E; Coronavirus OC43; Human Metapneumovirus; Human Rhinovirus/Enterovirus; Influenza A; Influenza A/H1; Influenza A/H3; Influenza A/H1-2009; Influenza B; Parainfluenza Virus 1/2/3/4; Respiratory Syncytial Virus BACTERIA:Bordetella parapertussis; Bordetella pertussis; Chlamydia pneumoniae; Mycoplasma pneumoniae * Available only on the BioFire Respiratory Panel 2	Real-time RT-PCR molecular test	nasopharyngeal swab in transport media	97.1%	99.3%	45 min
BIOFIRE	The BioFire^®^ FilmArray^®^ Respiratory EZ (RP EZ) Panel	VIRUSES:Adenovirus; Coronavirus; Human Metapneumovirus; Human Rhinovirus/Enterovirus; Influenza A; Influenza A/H1; Influenza A/H3; Influenza A/H1-2009; Influenza B; Parainfluenza Virus; Respiratory BACTERIA: Bordetella pertussis; Chlamydophila pneumoniae; Mycoplasma pneumoniae	Real-time RT-PCR molecular test	nasopharyngeal swab	BAL: 96.2% Sputum: 96.3%	BAL: 98.3% Sputum:97.2%	45 min
Cepheid	Xpert^®^ Xpress SARS-CoV-2	SARS-CoV-2	Real-time RT-PCR molecular test results	nasal swab, nasopharyngeal swab, aspirate specimens	100%	100%	45 min
Cepheid	Xpert^®^Xpress Flu/RSV	Detects viral RNA, enabling better detection of seasonal mutations of the flu virus (A,b,RSV)	Real-time RT-PCR molecular test results	nasal swab, nasopharyngeal swab specimens	Flu A: 98.1% Flu B: 100% RSV: 98.4%	Flu A: 98.1% Flu B: 99.1% RSV: 99.3%	20 min

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
