# Peer review of "Methods of Respiratory Virus Detection: Advances towards Point-of-Care for Early Intervention"

_micromachines, 2021, doi:10.3390/mi12060697_

Round 1

Reviewer 1 Report

  • I recommend to identify POC or POCT during the text for the correct context meaning and replace homogeneously
  • I suggest replace reference 3 by other most suitable about Covid-19 outbreak
  • It is most appropriate write IgG and IgM instead of IGG and IGM
  • I recommend to change the reference 7 for a review about Covid-19 diagnosis (immunological reaction and virus load)
  • In Table 1, it will be interesting for the reader to compare also the cost (approximately) of each diagnostic technique
  • As a keyword is better to write COVID-19 instead COVID-2019 or 2019-nCov. I recommend to identify and replace during the text
  • Is it possible to replace reference 12 and 13 for others more actual about this area?
  • In reference 29 the authors may add the link (web page)
  • Some write errors appear during the text, please it should be revised
  • Tittle 3. Commercialized respiratory virus diagnostics for POCT could be most appropriated as a 3. Commercialized respiratory virus diagnostic POCT devices
  • In part 3.2, line 129 it is recommended to re-write the phrase to clarify the justification/importance of the serological antibody testing
  • It could be interesting to add the % of false positive results in line 130-131
  • I recommend to rewrite or reconsider the phrase of the line 133-134
  • I recommend replace the subtitle 4.2. ELISA-based test strips, for another most precise about paper devices as for example "Immunoassay paper-based devices/technology"
  • In figure 2B(ii) the graphs are not clear. I recommend omit the figure 2B(ii) or only show the most representative graph
  • The reference 40 is correctly cited? What type of reference is?
  • In figure 4A(ii) and 4B(ii) the graphs are not clear. I recommend omit the figures or only show the most representative graphs
  • I suggest reconsider the phrase of the lines 275-277 as conclusion for the review

Author Response

Dear Editor,

Thank you very much for your time and effort reviewing our manuscript. We would also like to express our gratitude to the reviewers for their invaluable and constructive comments to help us significantly improve our manuscript. We have addressed their comments item-by-item in the response letter and revised the manuscript accordingly. Now, we strongly believe that the manuscript is of the quality, novelty and broad interest that would satisfy all the necessary criteria to be published in Micromachines.

For your convenience, the reviewers’ comments are shown in blue italic in the response letter, and corresponding changes are tracked and underlined in red in the revised manuscript.

Yours sincerely,

Liguo Liang, PhD.

The First Affiliated Hospital, Zhejiang University School of Medicine,

State Key Laboratory for Diagnosis and Treatment of Infectious Diseases,

Hangzhou, China, 310003

Email: lianglg@zju.edu.cn

Reviewers' comments:

Reviewer #1:

I recommend to identify POC or POCT during the text for the correct context meaning and replace homogeneously

Thanks for your comments. We deleted POCT and only the POC kept in the text. (Full text of this article, there are 18 places in total).

I suggest replace reference 3 by other most suitable about Covid-19 outbreak

Thangs for your comments. We replaced a new reference for COVID-19 outbreak. (page 2, line 24).

Zhao, H., COVID-19 drives new threat to bats in China. Science, 2020. 367(6485): p. 1436.

It is most appropriate write IgG and IgM instead of IGG and IGM

Thanks for your comments. We have revised it. (Full text of this article).

I recommend to change the reference 7 for a review about Covid-19 diagnosis (immunological reaction and virus load)

Thanks for your comments. We deleted this reference, and insert another two references, which about immune responses against SARS-CoV-2 and antibody testing. (page 2, line 34).

(Mahalingam, S., et al., Landscape of humoral immune responses against SARS-CoV-2 in patients with COVID-19 disease and the value of antibody testing. Heliyon, 2021. 7(4): p. e06836.

Long, Q.X., et al., Antibody responses to SARS-CoV-2 in patients with COVID-19. Nat Med, 2020. 26(6): p. 845-848.)

In Table 1, it will be interesting for the reader to compare also the cost (approximately) of each diagnostic technique

Thanks for your comments. We have revised Table 1.  Since health and routine checkups or testing of COVID with respect to cost is not the talk of the hour, we deleted it. (page 2, line 34).

As a keyword is better to write COVID-19 instead COVID-2019 or 2019-nCov. I recommend to identify and replace during the text

Thanks for your comments. We have revised it. (page 3, line 51-52).

Is it possible to replace reference 12 and 13 for others more actual about this area?

Thanks for your comments,we have replaced the new references about this area. (page 3, line 39 and line 53).

Arizti-Sanz, J., et al., Streamlined inactivation, amplification, and Cas13-based detection of SARS-CoV-2. Nat Commun, 2020. 11(1): p. 5921.

Leland, D.S. and C.C. Ginocchio, Role of cell culture for virus detection in the age of technology. Clin Microbiol Rev, 2007. 20(1): p. 49-78.

In reference 29 the authors may add the link (web page)

Thanks for your comments,we have added the link for reference 30 (Original reference 29). (page 4, line 87).

https://www.fiercebiotech.com/medtech/fda-grants-roche-coronavirus-test-emergency-green-light-within-24-hours

Some write errors appear during the text, please it should be revised

Thanks for your comments. We have gone through the whole text and revised them.

Tittle 3. Commercialized respiratory virus diagnostics for POCT could be most appropriated as a 3. Commercialized respiratory virus diagnostic POCT devices

Thanks for your comments. We have revised it. (page 5, line 94).

In part 3.2, line 129 it is recommended to re-write the phrase to clarify the justification/importance of the serological antibody testing

(page 7, line 132).

Thanks for your comments. We have revised it

“Therefore, serological antibody testing is important to verify the effectiveness of virus treatment.”

It could be interesting to add the % of false positive results in line 130-131

Thanks for your comments. There is no report for the % of FALSE positive results, so we have deleted the sentence which is inappropriate.

I recommend to rewrite or reconsider the phrase of the line 133-134

Thanks for your comments. We have revised it.

“During a virus outbreak, rapid detection of respiratory viral load is essential to facilitate decision making on the initiation of anti-respiratory viral treatment (ART) or the identification of virological ART failure.” (page 8, line 135).

I recommend replace the subtitle 4.2. ELISA-based test strips, for another most precise about paper devices as for example "Immunoassay paper-based devices/technology"

Thanks for your comments. We have revised it. (page 11, line 170).

In figure 2B(ii) the graphs are not clear. I recommend omit the figure 2B(ii) or only show the most representative graph

Thanks for your comments. We have revised figures. (page 12, line 190).

The reference 40 is correctly cited? What type of reference is?

Thanks for your comments. We have revised this reference.

(page 7, line 132).

In figure 4A(ii) and 4B(ii) the graphs are not clear. I recommend omit the figures or only show the most representative graphs

Thanks for your comments. We have revised figures. (page 17, line 258).

I suggest reconsider the phrase of the lines 275-277 as conclusion for the review

Thanks for your comments. We have revised it. (page 21, line 316-318).

“In the meantime, with the global outbreak of COVID-19 still underway, researchers should effectively apply new technologies (i.e., SERS, digital PCR, plasmonic-based biosensors) for rapid virus detection to improve detection sensitivity and avoid missing the detection of low-copy viral load”

Reviewer 2 Report

Methods of respiratory virus detection: Advances towards point-of-care for early intervention

Siming Lu2, Sha Lin2, Hongrui Zhang1, Liguo Liang3* and Shien Shen1*

Comments and Suggestions for Authors

In this review paper the authors documented recent advances in point-of-care (POC) techniques for detections and early interventions of respiratory disease causing virus. They covered a wide spectrum of POC techniques ranging from conventional immunofluorescence assay (IFA) to recently developed microfluidics and smart-phone based techniques. Overall, this is a very good piece of review article on detection of respiratory viruses in current time, especially during the covid-19 pandemic. I would like to recommend accepting the review paper after minor revisions. Following are the comments for minor revisions to the authors.

  1. Along with the POC-techniques mentioned in the review, there are many promising ones having potential to be adopted as future POC diagnosis tools for detection of respiratory viruses. For example, digital PCR, surface-enhanced Raman scattering (SERS)-based lateral flow assay (LFA) paper-strips, 3D plasmonic chips ( Chen et al. Biosensors and Bioelectronics 167 (2020) 112496) etc. have big potential to be used as POC techniques in recent future due to their high sensitivity for the target virus. It is recommended to include a section as ‘Potential techniques for future POC diagnoses’ to this review article.
  2. In 4.1 Rapid molecular diagnosing section, line 173, i.e., ‘…such as gold nanoparticles (AuNPs), have been increasingly employed in the development of lateral flow test strips for their unique optical, electronic, and/or structural properties.’ demands references for each properties of gold nanoparticles mentioned. It is needed to add a very unique and useful property of gold nanoparticles, i.e., plasmonic property ( Das et al. J Raman Spectrosc. 2021;52:506–515)
  3. Descriptions of the figures (i) and (ii) of Figure 1 need to be elaborated.
  4. Use ‘COVID-19’ instead of ‘COVID-2019’
  5. Remove ‘(LAMP)’ from the heading of Section ‘4.1 Rapid molecular diagnosis (LAMP)’.
  6. If possible, improve English language of the manuscript.

Author Response

Dear Editor,

Thank you very much for your time and effort reviewing our manuscript. We would also like to express our gratitude to the reviewers for their invaluable and constructive comments to help us significantly improve our manuscript. We have addressed their comments item-by-item in the response letter and revised the manuscript accordingly. Now, we strongly believe that the manuscript is of the quality, novelty and broad interest that would satisfy all the necessary criteria to be published in Micromachines.

For your convenience, the reviewers’ comments are shown in blue italic in the response letter, and corresponding changes are tracked and underlined in red in the revised manuscript.

Yours sincerely,

Liguo Liang, PhD.

The First Affiliated Hospital, Zhejiang University School of Medicine,

State Key Laboratory for Diagnosis and Treatment of Infectious Diseases,

Hangzhou, China, 310003

Email: lianglg@zju.edu.cn

Reviewer #2:

In this review paper the authors documented recent advances in point-of-care (POC) techniques for detections and early interventions of respiratory disease-causing virus. They covered a wide spectrum of POC techniques ranging from conventional immunofluorescence assay (IFA) to recently developed microfluidics and smart-phone based techniques. Overall, this is a very good piece of review article on detection of respiratory viruses in current time, especially during the covid-19 pandemic. I would like to recommend accepting the review paper after minor revisions. Following are the comments for minor revisions to the authors.

  1. Along with the POC-techniques mentioned in the review, there are many promising ones having potential to be adopted as future POC diagnosis tools for detection of respiratory viruses. For example, digital PCR, surface-enhanced Raman scattering (SERS)-based lateral flow assay (LFA) paper-strips, 3D plasmonic chips (Chen et al. Biosensors and Bioelectronics 167 (2020) 112496) etc. have big potential to be used as POC techniques in recent future due to their high sensitivity for the target virus. It is recommended to include a section as ‘Potential techniques for future POC diagnoses’ to this review article.

Thanks for your comments. We added the new part of in the review article as ‘Potential techniques for future POC diagnoses’ (page 20, line 281-307).

Potential techniques for future POC diagnosis

In addition to the new technologies listed above, there are many promising ones having potential to be adopted as future POC diagnosis tools for detection of respiratory viruses, such as digital PCR, surface-enhanced Raman scattering (SERS), 3D plasmonic-based chips. For example, rapid detection of low virus (SARS-CoV-2) RNA load was achieved by using a droplet digital PCR method[1]. In this study, a flat pipette head with an elliptical cross section was utilized to prepare monodisperse droplets. By the end of PCR reaction, the fluorescence generated by the droplets with virus (as low as 100 target copies) can be captured by the mobile phone software, and the presence or absence of the virus can be identified by naked eyes or images. This platform demonstrated a LODs of 3.8 copies per 20 μL of sample with a dynamic range of 4-100 copies. Additionally, the ddPCR platform is shown to be inhibitor resistant with spiked saliva samples, indicated that RNA extraction may not be necessary.

Surface-enhanced Raman scattering (SERS) was reported to improve the sensitivity of detection of influenza A/H1N1 virus [2]. In this study, the surface energy difference between a perfluoro decanethiol (PFDT) spacer and the Au layer was utilized to fabricate a 3D nano-popcorn plasmonic substrate, making gold nanoparticles self-assemble and generating multiple hot spots between neighboring particles. The decrease of Raman peak intensity resulting from the release of Cy3-labeled aptamer DNAs from nano-popcorn substrate surfaces via the interaction between the aptamer DNA and A/H1N1 virus was used to quantitate the influenza A/H1N1 virusThis method showed approximately three orders of magnitude more sensitive than that determined by the ELISA assay.

The plasmonic model, which can enhance the performance of existing platforms by providing stable, real-time, highly sensitive and label-free analyte detection, has emerged as a key candidate for the development of next-generation diagnostic technologies to reduce the burden of infectious diseases [3, 4]. For example, a dual-functional plasmonic biosensor combining the plasmonic photothermal (PPT) effect and localized surface plasmon resonance (LSPR) sensing transduction was fabricated for COVID-19 diagnosis [5]. In the biosensor, the gold nanoislands (AuNIs) functionalized with complementary DNA receptors can perform a sensitive detection of the selected sequences from severe acute respiratory syndrome coronavirus 2 (SARS-CoV-2) through nucleic acid hybridization. For better sensing performance, the thermoplasmonic heat was generated on the same AuNIs chip when illuminated at their plasmonic resonance frequency. The localized PPT heat was capable to elevate the in-situ hybridization temperature and facilitate the accurate discrimination of two similar gene sequences. This biosensor exhibited a high sensitivity toward the selected SARS-CoV-2 sequences with a lower detection limit down to the concentration of 0.22 pM and allows precise detection of the specific target in a multigene mixture. These studies reveal the applicability of digital PCR, surface-enhanced Raman scattering (SERS), plasmonic-based biosensors and other technologies in nucleic acid testing and viral disease diagnosis, and provide an effective idea for the rapid detection of respiratory viruses in the future.

  1. Chen, L., et al., Elliptical Pipette Generated Large Microdroplets for POC Visual ddPCR Quantification of Low Viral Load. Anal Chem, 2021. 93(16): p. 6456-6462.
  2. Chen, H., et al., SERS imaging-based aptasensor for ultrasensitive and reproducible detection of influenza virus A. Biosens Bioelectron, 2020. 167: p. 112496.
  3. Li, Z., et al., Plasmonic-based platforms for diagnosis of infectious diseases at the point-of-care. Biotechnol Adv, 2019. 37(8): p. 107440.
  4. Soler, M., et al., How Nanophotonic Label-Free Biosensors Can Contribute to Rapid and Massive Diagnostics of Respiratory Virus Infections: COVID-19 Case. ACS Sens, 2020. 5(9): p. 2663-2678.
  5. Qiu, G., et al., Dual-Functional Plasmonic Photothermal Biosensors for Highly Accurate Severe Acute Respiratory Syndrome Coronavirus 2 Detection. ACS nano, 2020.

  1. In 4.1 Rapid molecular diagnosing section, line 173, i.e., ‘…such as gold nanoparticles (AuNPs), have been increasingly employed in the development of lateral flow test strips for their unique optical, electronic, and/or structural properties.’ demands references for each properties of gold nanoparticles mentioned. It is needed to add a very unique and useful property of gold nanoparticles, i.e., plasmonic property (Das et al. J Raman Spectrosc. 2021; 52:506–515)

Thanks for your comments. We have added the reference as list reference 45(page 11, line177-180).

For example, Das et al, reported a rapid and sensitive determination of the total iron-binding capacity of transferrin (Tf) in human serum using surface-enhanced Raman scattering (SERS) spectroscopy[45]. In the study, metal organic framework (MOF)–gold nanoparticle (AuNP) complexes were used as the SERS substrate. This SERS-based analysis of Tf using MOF–AuNPs provides new insight for the rapid and sensitive diagnosis of iron deficiency in human serum.

  1. Descriptions of the figures (i) and (ii) of Figure 1 need to be elaborated.

Thanks for your comments. We have revised this caption. (page 11, line 167-169).

  1. Use ‘COVID-19’ instead of ‘COVID-2019’

Thanks for your comments. We have revised it.

  1. Remove ‘(LAMP)’ from the heading of Section ‘4.1 Rapid molecular diagnosis (LAMP)’.

Thanks for your comments. We have revised it. (page 8, line 144).

  1. If possible, improve English language of the manuscript.

Thanks for your comments. We have gone through the whole text and revised the write errors.

Reviewer 3 Report

The review paper “Methods of respiratory virus detection: Advances towards point-of-care for early intervention” by Siming Lu et.al., summarizes the conventional and emerging  methods in respiratory virus detection and outlines the importance of point-of-care (POC) technologies in monitoring/detecting respiratory virus. The concept and idea of the work is good, and since its’s a hot topic sharing and presenting information will gather a lot of visibility. However, sharing the right information is imperative to scientific community as well as to align with the scope of the journal. This manuscript needs major revisions, only then I can make a decision.

  1. Kindly check grammar and the English paraphrase sentences throughout.
  2. Line 12, “ However, Current methods for, ……………………... and are time-consuming. The ‘C’ in current should be lowercase.
  3. Line 13, “Also, for antibodies detection …..or viral control” . Kindly paraphrase. The sentence does not make any sense.
  4. Line 15, “LAMP”. Full form please.
  5. Line 22, “wellbeing” to well-being
  6. Line 34,36, “(IGG&IGM),” to (IgG&IgM). Kindly change throughout.
  7. Line 41-42, Therefore, there is an urgent need for viral load detection to manage infected patients and thus effectively control the spread of the virus in time. This is an absurd statement. Kindly rethink what has been written. I believe here the idea is that these technologies can help monitor the virus or detect the virus and provide information in a timely manner. “Control the spread of the virus in time” is not in the scope of this paper.
  8. Line 50, “or sequencing are rare in clinical laboratories”. Kindly replace it with better statement like for example, “or sequencing are limited in clinical laboratories”
  9. Table 1. This is very questionable. All these methods are well established. However, no assay/method gives 100 % repeatability. Kindly delete this section as it is not true. Also, Workload column is unnecessary. Cost is also a very questionable. Based on my understanding, health and routine checkups or testing of COVID with respect to cost is not the talk of the hour. The need at current and I believe the scope of this work is to present and outline conventional and emerging methods in respiratory virus detection.   
  10. Line 117, Kindly paraphrase to a more meaningful sentence.
  11. Line 133-134 , “anti-respiratory viral treatment (ART) or the identification of virological ART failure” Kindly un-bold.
  12. Section 4.1 onwards needs major revision.
  13. Figure 1, “LAMP-VF”. Nowhere its mentioned that VF means Vertical Flow. Kindly add. The figures must be self-explanatory and not just a copy paste from literature or source. (ii) Comparison … I can see no comparison. Kindly address that one is gel electrophoresis, and another is vertical flow. Make figures look appealing and presentable.  
  14. Line 166, “Reference[42]”. Kindly write :Reprinted with permission from [42]. Copyright permissions should be addressed throughout. Kindly refer journal guidelines.
  15. Figure 2-5. Kindly present it in a way that words/texts can be seen. I had a hard time reading.

Kindly understand that collecting information is not “Review” paper/article. Drafting the manuscript in a readable way is very important for a good review paper. Kindly put some time to re-address all the figures in a presentable manner rather than just throwing in a bunch of figures which cannot be understood at all. This article has a lot of current information and if addressed and presented in a readable format, I can re-visit on my decision.

Author Response

Reviewer 3

The review paper “Methods of respiratory virus detection: Advances towards point-of-care for early intervention” by Siming Lu et.al., summarizes the conventional and emerging methods in respiratory virus detection and outlines the importance of point-of-care (POC) technologies in monitoring/detecting respiratory virus. The concept and idea of the work is good, and since its’s a hot topic sharing and presenting information will gather a lot of visibility. However, sharing the right information is imperative to scientific community as well as to align with the scope of the journal. This manuscript needs major revisions, only then I can make a decision.

  1. Kindly check grammar and the English paraphrase sentences throughout.

Thanks for your comments. We have gone through the whole text and revised the write errors.

  1. Line 12, “However, Current methods for, ……………………... and are time-consuming. The ‘C’ in current should be lowercase.

Thanks for your comments. We have revised it. (page 1, line 14).

  1. Line 13, “Also, for antibodies detection …..or viral control” . Kindly paraphrase. The sentence does not make any sense.

(page 1, line 15-17).

Also, for COVID-19 detection is generally relied on an automated PCR analyzer, which is highly instrument-dependent and expensive. As such, emerging technologies in the development of respiratory viral load assays for point-of-care (POC) testing is urgently needed for viral screening.

  1. Line 15, “LAMP”. Full form please.

Thanks for your comments. We have revised it. (page 1, line 17).

  1. Line 22, “wellbeing” to well-being

Thanks for your comments. We have revised it. (page 1, line 23).

  1. Line 34,36, “(IGG&IGM),” to (IgG&IgM). Kindly change throughout.

Thanks for your comments. We have revised it. (page 1, line 33).

  1. Line 41-42, Therefore, there is an urgent need for viral load detection to manage infected patients and thus effectively control the spread of the virus in time. This is an absurd statement. Kindly rethink what has been written. I believe here the idea is that these technologies can help monitor the virus or detect the virus and provide information in a timely manner. “Control the spread of the virus in time” is not in the scope of this paper.

Thanks for your comments. We have revised it. For your convenient, we put this sentence below:

“Therefore, there is an urgent need for viral load detection to monitor the virus and thus provide information to manage infected patients in a timely manner.” (page 3, line 40-41).

  1. Line 50, “or sequencing are rare in clinical laboratories”. Kindly replace it with better statement like for example, “or sequencing are limited in clinical laboratories”

Thanks for your comments. We have revised it. (page 3, line 49).

  1. Table 1. This is very questionable. All these methods are well established. However, no assay/method gives 100 % repeatability. Kindly delete this section as it is not true. Also, Workload column is unnecessary. Cost is also a very questionable. Based on my understanding, health and routine checkups or testing of COVID with respect to cost is not the talk of the hour. The need at current and I believe the scope of this work is to present and outline conventional and emerging methods in respiratory virus detection.  

Thanks for your comments. We have revised it.  (page 3, line 51-52).

  1. Line 117, Kindly paraphrase to a more meaningful sentence.

Thanks for your comments. We have revised this sentence and described more about Xpert® Xpress SARS-CoV-2.

“Recently, as COVID-19 has greatly dispersed around the world, Xpert® Xpress SARS-CoV-2 was fabricated for the rapid detection of SARS‑CoV‑2, which have been demonstrated to be highly-sensitive tests in numerous analytical and clinical studies [36, 37]. This kit provides the speedy detection of this virus in 45 minutes, and it only takes one minute to complete the sample preparation. Also, this kit showed sensitivity of 100% with a limit of detection (LOD) ranging from 8.3 to 60 cp/mL for SARS-CoV-2 detection. As a rapid detection device, this kit showed excellent agreement with the Roche Cobas 6800 system and traditional RT-qPCR.”   (page 7, line 115-119).

  1. Line 133-134, “anti-respiratory viral treatment (ART) or the identification of virological ART failure” Kindly un-bold.

Thanks for your comments. We have revised it. (page 8, line 135-136).

  1. Section 4.1 onwards needs major revision.

Thanks for your comments. We have revised it. (page 8, line 144-160;page 9,line 161-165).

4.1. Rapid molecular diagnosis

Loop-mediated isothermal amplification (LAMP) is a rapid molecular diagnosis based in a novel nucleic acid amplification method, was initially developed by Notomi et al [41]. It relies on 4 primers (2 outer primers and 2 inner primers) and a strand displacement DNA polymerase. The 4 primers can recognize 6 specific fragments of conserved-sequence DNA. The main principle of LAMP technology is that DNA can be in a dynamic equilibrium state at about 65℃. At this temperature, 4 specific primers and a strand displacement DNA polymerase are utilized to synthesize DNA by self-circulating. Based on this method, several infection kits have been developed for POC. For example, in November 2020, FDA issued an emergency authorization to approve the Lucira Covid-19 All-in-One Test Kit — the first home self-test kit that could provide rapid results [42]. The kit utilizes LAMP technology to rapidly produce results within 30 minutes and has a 100% conformance rate compared to highly sensitive detection methods. Furthermore, based on the characteristics of LAMP and using its principle, many studies have been dedicated to the development of rapid detection methods for respiratory viruses.

Recently, researchers have developed LAMP-based novel assays to detect respiratory viruses, such as influenza viruses, MERS-CoV, and SARS-CoV, for POC testing. For example, a rapid and colorimetric multiplex reverse transcription LAMP (RT-LAMP) method was developed for influenza viruses (e.g., H5N1, H5N6, H1N1, H3N2 and H7N9) [43]. Compared to conventional PCR methods, which need 2~3 h for the full reaction and observation, this novel method only requires 1 h for several target genes (i.e., H1N1, H3N2, H5N1, H5N6, H5N8 and H7N9) and without additional steps. Moreover, the RT-LAMP assay shows specificity to infectious viruses and capability to intuitively detect these viruses by a one-pot colorimetric visualization assay, making it a more feasible POC test. This test showed a high sensitivity of 100 to 0.1 viral genome copies. In another study, a nucleic acid visualization method was developed for detecting MERS-CoV nucleic acids, which is based on a vertical-flow colloidal gold particle binding strip (Figure 1) [44]. This novel visualization method shows significant advantages over existing MERS-CoV rapid detection assays, since two loop primers (LF and LB) are labeled with FITC or biotin to provide visible results. This assay was shown to be capable of detecting 1 × 101 copies/µL of MERS-CoV RNA, which is regarded as more sensitive than a conventional RT-LAMP assay. In addition, it showed high specificity and exhibited no cross-reactivity with any of SARS-CoV, HKU4, HKU1, OC43 and 229E. Considering the equipment-free nature and rapid result output, the visualized strip is a promising approach for the management of MERS at the POC approach. Focusing on the currently circulating SARS-CoV-2, LAMP is expected to play an important role in its detection, so that infected persons will be localized as soon as possible for isolation to avoid the further spread of the virus and thus slow down the outbreak.

  1. Figure 1, “LAMP-VF”. Nowhere its mentioned that VF means Vertical Flow. Kindly add. The figures must be self-explanatory and not just a copy paste from literature or source. (ii) Comparison … I can see no comparison. Kindly address that one is gel electrophoresis, and another is vertical flow. Make figures look appealing and presentable.

Thanks for your comments. We have revised it, and make figures look appealing.  

  1. Line 166, “Reference [42]”. Kindly write: Reprinted with permission from [42]. Copyright permissions should be addressed throughout. Kindly refer journal guidelines.

Thanks for your comments. We have written copyright permissions.

  1. Figure 2-5. Kindly present it in a way that words/texts can be seen. I had a hard time reading.

Kindly understand that collecting information is not “Review” paper/article. Drafting the manuscript in a readable way is very important for a good review paper. Kindly put some time to re-address all the figures in a presentable manner rather than just throwing in a bunch of figures which cannot be understood at all. This article has a lot of current information and if addressed and presented in a readable format, I can re-visit on my decision.

Thanks for your comments. We have revised all figures.

Round 2

Reviewer 3 Report

No comments